# Large Scale Process for Low Crystalline MoO_3_-Carbon Composite Microspheres Prepared by One-Step Spray Pyrolysis for Anodes in Lithium-Ion Batteries

**DOI:** 10.3390/nano9040539

**Published:** 2019-04-03

**Authors:** Jung Sang Cho

**Affiliations:** Department of Engineering Chemistry, Chungbuk National University, Chungbuk 361-763, Korea; jscho@cbnu.ac.kr; Tel.: +82-43-261-2489; Fax: +82-43-262-2380

**Keywords:** molybdenum oxide, carbon composite, spray pyrolysis, anodes, lithium-ion batteries

## Abstract

This paper introduces a large-scale and facile method for synthesizing low crystalline MoO_3_/carbon composite microspheres, in which MoO_3_ nanocrystals are distributed homogeneously in the amorphous carbon matrix, directly by a one-step spray pyrolysis. The MoO_3_/carbon composite microspheres with mean diameters of 0.7 µm were directly formed from one droplet by a series of drying, decomposition, and crystalizing inside the hot-wall reactor within six seconds. The MoO_3_/carbon composite microspheres had high specific discharge capacities of 811 mA h g^−1^ after 100 cycles, even at a high current density of 1.0 A g^−1^ when applied as anode materials for lithium-ion batteries. The MoO_3_/carbon composite microspheres had final discharge capacities of 999, 875, 716, and 467 mA h g^−1^ at current densities of 0.5, 1.5, 3.0, and 5.0 A g^−1^, respectively. MoO_3_/carbon composite microspheres provide better Li-ion storage than do bare MoO_3_ powders because of their high structural stability and electrical conductivity.

## 1. Introduction

Lithium-ion batteries (LIBs) have been attractive as the most important type of power source for energy-storage system, electric vehicles, and other electronic devices because of their high specific capacities and energy densities [1,2,3]. Transition-metal oxides (TMOs) with high theoretical energy capacities have been widely applied as replacement anodes for the current graphite of LIBs [4,5,6]. However, the low intrinsic electric conductivity and the large volume expansion of TMOs during a charge/discharge cycle result in rapid capacity fading, which hinders the commercial application of TMOs for anodes in current LIBs [7,8]. To solve these problems, compositing TMOs with carbonaceous materials has been regarded as a possible solution. Carbon could effectively buffer the stress induced by the large volume change of TMOs during the fast charging–discharging process and improve the electrical conductivity of the anodes [9,10,11]. Additionally, a carbon matrix could prevent the aggregation of the active materials during repeated cycles by surrounding them, which increases the structural stability of anode materials [12,13]. Therefore, various synthesis strategies for TMOs/carbon composites have been introduced [14,15,16,17,18]. Cho et al. [14] prepared multiroom-structured metal–carbon hybrid microspheres containing empty voids of several tens of nanometers by liquid–liquid phase segregation because of the incongruent melting of the metal salt and dextrin during the spray pyrolysis. The discharge capacity of the multiroom-structured Co_3_O_4_–C hybrid microspheres for LIBs at a current density of 3 A g^−1^ for the 150th cycle was 1243 mA h g^−1^. Zhang et al. [15] also prepared TiO_2_–graphene composite nanofibers by a simple electrospinning process. The cell assembled with TiO_2_–graphene composite nanofibers as an anode retained 84% of the reversible capacity after 300 cycles at a current density of 150 mA g^−1^, which is 25% higher than bare TiO_2_ nanofibers did under the same test conditions. Bhaskar et al. [16] prepared MoO_2_/multiwalled carbon nanotubes (MWCNTs) composed of spherical flowerlike nanostructures of MoO_2_, interconnected by MWCNTs by a one-step hydrothermal route. The one-dimensional electron-transport pathways provided by MWCNTs, which are in direct contact with the MoO_2_ nanostructures, imparted an improved reversible lithium storage capacity (1143 mA h g^−1^ at a current density of 100 mA g^−1^ after 200 cycles).

Molybdenum oxides are candidate anode materials for LIBs because MoO_3_ exhibits good electrochemical properties, has a low cost, and is environmentally friendly [19,20,21]. Therefore, MoO_3_ nanomaterials with diverse morphologies such as nanoparticles, hollow, nanobelts, naowiles, and porous structures have been prepared. Lee et al. [22] synthesized MoO_3_ nanoparticles using hot filament chemical vapor deposition method (HFCVD) under an argon atmosphere. Zhao et al. [23] also synthesized MoO_3_ hollow microspheres by a template-free solvothermal route and subsequent heat treatment in air. The MoO_3_ hollow microspheres have a relatively high specific surface area. Chen et al. [24] prepared MoO_3_ nanobelts by a hydrothermal method, in which the morphology of MoO_3_ nanobelts was affected with the addition of PEG. MoO_3−x_ nanowires were prepared by Sunkara et al. [25] in a hot-filament chemical vapor deposition reactor. Ko et al. [26] prepared three-dimensional ordered macroporous structured MoO_3_ by using a polystyrene bead template via ultrasonic spray pyrolysis.

In this study, low crystalline MoO_3_/carbon composite microspheres, in which MoO_3_ nanocrystals were distributed homogeneously in the amorphous C matrix, were directly prepared by one-step spray pyrolysis within several seconds. In here, MoO_3_ was applied as the host material of carbon microspheres in this process because of its rich chemistry with multiple valence states, low electrical resistivity, high electrochemical activity toward lithium, and affordable cost. The resulting MoO_3_/carbon composite microspheres worked better in terms of cycling and rate as anode materials for LIBs than did bare MoO_3_ powders. The simple process introduced in this study is expected to be useful for the large-scale synthesis of TMOs/carbon composite microspheres as practical anode materials for LIBs. Furthermore, the synthesis strategy introduced is generally applied to synthesize various metal TMOs/carbon composites, including NiO, Co_3_O_4_, SnO_2_, and Fe_2_O_3_, for a wide variety of applications including energy storage.

## 2. Materials and Methods

### 2.1. Sample Preparation

Low crystalline MoO_3_/carbon composite microspheres, in which MoO_3_ nanocrystals were distributed homogeneously in the amorphous C matrix, were directly prepared by a one-step spray pyrolysis. The spray pyrolysis system used in this study is shown in Appendix A. In brief, droplets were generated by a 1.7-MHz ultrasonic spray generator that consisted of six vibrators, and the droplets were carried to a quartz tube reactor (length = 1200 mm, diameter = 50 mm) by a flow of N_2_ (flow rate = 5 L min^−1^). The reactor temperature was fixed at 900 °C. The spray solution was prepared by dissolving 0.1 M of MoO_3_ (98%, Sigma Aldrich, St. Louis, MO, USA), 12 g L^−1^ of polyvinylpyrrolidone (PVP, Mw 40,000, Daejung Chemicals and Metals, Siheung, Korea), and 0.02 M of sucrose in distilled water. Subsequently, an appropriate amount of hydrogen peroxide (30% H_2_O_2_, Sigma-Aldrich) was added to the above solution to obtain a clear spray solution. For the bare sample, MoO_3_ powders without any carbon content were also prepared by spray pyrolysis. For this, the spray solution was prepared by dissolving 0.1 M of MoO_3_ without a carbon precursor in H_2_O_2_ contained in distilled water. Subsequently, the spray pyrolysis was carried out with the prepared solution at a temperature of 900 °C by a flow of air (flow rate = 5 L min^−1^).

### 2.2. Characterization Techniques

The microstructures of the resulting powders were observed by scanning electron microscopy (SEM; JEOL, JSM-6060, JEOL, Tokyo, Japan) and field-emission transmission electron microscopy (FE-TEM; JEOL, JEM-2100F, JEOL, Tokyo, Japan). The crystal phases were evaluated by X-ray diffractometry (XRD; X’Pert PRO MPD, PANalytical, Almelo, The Netherlands) using Cu K_α_ radiation (λ = 1.5418 Å). X-ray photoelectron spectroscopy (XPS; K-Alpha, Thermo Fisher Scientific, Waltham, MA, USA) with a focused monochromatic Al K_α_ at 12 kV and 20 mA was used to analyze the composition of the samples. A structural characterization of carbon in the sample was performed by Raman spectra (Jobin Yvon LabRam HR800, Horiba Jobin Yvon, Paris, France, excited by a 632.8 nm He–Ne laser) at room temperature. The surface areas of the powders were measured by the Brunauer–Emmett–Teller (BET) method, using N_2_ as the adsorbate gas. Thermogravimetric analyses (TGA) were performed using a Pyris 1 TGA (Perkin Elmer, Waltham, MA, USA) within a temperature range of 25–650 °C and at a heating rate of 10 °C min^−1^ under a static air atmosphere.

### 2.3. Electrochemical Measurements

The electrochemical properties of the samples were analyzed by constructing a 2032-type coin cell. The lithium cell assembly was made in an Ar-filled glove box at room temperature where water and the oxygen concentration was kept at less than 1 ppm. The anode slurry was prepared by mixing the active material, carbon black, and sodium carboxymethyl cellulose (CMC) in a weight ratio of 7:2:1. The working electrodes were formed by coating the slurry onto copper foils and subsequently dried at 70 °C for 3 h. Li metal and a microporous polypropylene film were used as the counter electrode and the separator, respectively. The electrolyte was composed of 1 M LiPF_6_ dissolved in a mixture of fluoroethylene carbonate/dimethyl carbonate (FEC/DMC; 1:1 *v*/*v*). The discharge/charge characteristics of the samples were investigated by cycling over a potential range of 0.001–3.0 V under CC (constant-current) conditions. Cyclic voltammograms were measured at a scan rate of 0.1 mV s^−1^. The negative electrode measured 1.5 cm × 1.5 cm, and the mass loading of the active materials was kept at approximately 1.5 mg cm^−2^ in every electrochemical test. The electrochemical impedance spectra were obtained by performing alternating current electrochemical impedance spectroscopy (EIS; ZIVE SP1) over a frequency range of 0.01 Hz to 100 kHz.

## 3. Results and Discussion

Low crystalline MoO_3_/C composite microspheres, in which MoO_3_ nanocrystals were distributed homogeneously in the amorphous C matrix, were directly prepared by a one-step spray pyrolysis without any further treatment. Figure 1 shows the morphologies of the MoO_3_/C composite microspheres obtained after the one-step spray pyrolysis. The powders were spherical and had diameters on the order of microns because they were formed from one droplet with several tens of micrometers by drying, decomposition, and crystallization inside the hot-wall reactor, as shown in Scheme 1. Additionally, there was no aggregation between the powders because the spray pyrolysis was carried out within a very short residence time of 6 s in a hot-wall reactor maintained at 900 °C under a N_2_ atmosphere in Figure 1a,b. From a high-resolution TEM image in Figure 1c, it was hard to confirm the nanocrystal MoO_3_ grains formed during spray pyrolysis in a microsphere structure because the amorphous-like, very small MoO_3_ nanocrystals were formed during the spray pyrolysis at 900 °C within a short residence time of 6 s. The XRD result also showed the broad peak intensities of the β-MoO_3_ phase in Figure 1d. The mean crystallite size of the MoO_3_ powders, which was calculated from the width of the (011) peak using Scherrer’s equation, was 4 nm. Grain growth of the MoO_3_ nanocrystals was effectively prohibited both by the short residence time of the droplet in the reactor and by being surrounded by the carbon formed by the decomposition of PVP and sucrose during the process. The elemental mapping images shown in Figure 1e exhibited a homogeneous distribution of Mo, O, and C, which implies that the ultrafine MoO_3_ nanocrystals were homogeneously composited with C in the microsphere structure.

To identify the chemical composition of the MoO_3_/C composite microspheres, XPS analysis was carried out, as shown in Figure 2. The XPS survey spectrum of the composite microspheres confirmed the presence of Mo, O, and C, as shown in Figure 2a. In the Mo 3d spectrum of the microspheres (Figure 2b), the main peaks occurred at binding energies of 231.7/232.7 eV for Mo 3d5/2 and 234.7/235.7 eV for Mo 3d3/2; the peaks located at 232.7 and 235.7 eV are characteristic of typical values of the 3d orbital doublet Mo^6+^, and the minor ones centered on 231.7 and 234.7 eV corresponded to the 3d orbital doublet Mo^5+^, which indicated that dangling bond sites where charges could be trapped existed in MoO_3_ [27,28]. The C 1s XPS peak observed at 284.6 eV in Figure 2d corresponds to the binding energy of the sp^2^ C–C bond of the carbon matrix [29,30,31].

The carbon matrix of the MoO_3_/C composite microspheres was characterized by means of Raman spectroscopy. The degree of graphitization of the carbon material can typically be evaluated according to the intensity ratio of the D and G bands of carbon at approximately 1350 and 1590 cm^−1^, respectively [32,33]. The peak intensity ratio between the D and G bands (I_D_/I_G_) for the MoO_3_/C composite microspheres was approximately 3.2, and the absence of the 2D band at approx. 2685 cm^−1^ demonstrated that the carbon formed in the composite was fairly disordered. Thus, a large amount of the amorphous carbon was formed by the decomposition of both PVP and sucrose during the spray pyrolysis. In general, amorphous carbon has more capacity as an anode for LIBs than graphitic carbon, which is mainly contributed by pores and voids in the microcavities of the structure. The Thermogravimetric (TG) curve of the MoO_3_/C composite microspheres in Figure 3b revealed a weight loss between 380 and 460 °C because of the degradation of amorphous carbon. Therefore, the content of amorphous carbon of the MoO_3_/C composite microspheres estimated from the TG analysis was 26 wt %.

In order to clearly prove the structural merits of MoO_3_/C composite microspheres as anodes for Li^+^ ion storage properties, bare MoO_3_ powders without C were also prepared from the spray solution without either PVP and sucrose by spray pyrolysis, as shown in Figure 4. The mean particle size of the resulting bare MoO_3_ powders, as measured from the SEM and TEM images in Figure 4a,b, was 420 nm and had no aggregation between the powders. Additionally, the resulting powders were angular, which is attributed to the crystal growth of MoO_3_ particles because there was no carbon surrounding the particles during spray pyrolysis to prevent the growth of MoO_3_ crystals during the short residence reaction time of the droplets. The high-resolution TEM image in Figure 4c shows clear lattice fringes separated by 0.23 nm, which corresponds to the (011) crystal plane of β-MoO_3_ (JCPDS card No. 37–1445) [34]. The XRD pattern of the bare MoO_3_ powders (Figure 4d) shows that they have different allotropes of MoO_3_ structures, with no impurities. The thermodynamically favored α-MoO_3_ phase was newly formed along with β-MoO_3_ in the bare MoO_3_ powders during spray pyrolysis. Bare MoO_3_ powders without C were further confirmed by the elemental mapping images in Figure 4e. The BET surface areas of the MoO_3_/C composite microspheres and of the bare MoO_3_ powders were 4.3 and 0.6 m^2^ g^−1^, respectively, in Appendix A.

The electrochemical properties of the MoO_3_/C composite microspheres are compared with those of the bare MoO_3_ powders in Figure 5. The cyclic voltammogram (CV) curves of the MoO_3_/C composite microspheres and bare MoO_3_ powders performed in the 0.01–3.0 V range at a scanning rate of 0.01 mV s^−1^ for the first four cycles are shown in Figure 5a. In the first cathodic scan of the MoO_3_/C composite microspheres, the broad peaks located at 1.16 V and 0.21 V are assigned to the interaction of Li^+^ ions with the amorphous carbon matrix of the MoO_3_/C composite and conversion reaction of Li*_x_*MoO_3_ to Mo_0_ and Li_2_O [35,36,37]. The peak at 0.05 V is also observed, caused by the Li^+^ ion’s intercalation into the C matrix [38,39]. In the anodic scans of the MoO_3_/C composite microspheres, reversible peaks at 1.42 and 1.77 V are attributed to the monoclinic-orthorhombic-monoclinic phase transitions in the partially lithiated Li*_x_*MoO_2_ [35,36,37]. In the subsequent cycles, two redox peak pairs appeared at 0.21/1.3 and 1.42/1.77 V, which corresponded to the redox reaction of MoO_3_ [35,40,41]. The bare MoO_3_ powders showed peaks at 2.03 and 1.8 V in the first cathodic scan, which correspond to the generation of Li*_x_*MoO_3_, causing an irreversible structural change from the α-MoO_3_ additionally formed in the bare MoO_3_ powders to an amorphous phase [40,41,42]. The subsequent peak at 0.17 V results from the conversion reaction of Li*_x_*MoO_3_ to Mo_0_ and Li_2_O [35,36,37,41].

The initial discharge-charge curves of the two samples at a current density of 1.0 A g^−1^ are shown in Figure 5b. The initial discharge capacities of the MoO_3_/C composite microspheres and the bare MoO_3_ powders were 1403 mA h g^−1^ and 1478 mA h g^−1^, respectively, and their initial Coulombic efficiencies were 75% and 72%, respectively. Although the MoO_3_/C composite microspheres contained C with a high irreversible capacity loss, the initial Coulombic efficiency of the MoO_3_/C composite microspheres was relatively higher than that of the bare MoO_3_ powders. The high structural damage to the bare MoO_3_ powders in the first discharge and charge processes resulted in a low initial Coulombic efficiency. The discharge capacity and cycling properties of the MoO_3_/C composite microspheres and bare MoO_3_ powders at a current density of 1.0 A g^−1^ are shown in Figure 5c. Compared with bare MoO_3_ powders, the MoO_3_/C composite microspheres exhibited a satisfactorily stable cycling performance. The discharge capacity of the MoO_3_/C composite microspheres decreased slightly from 1066 mA h g^−1^ (533 mA h cc^−1^) to 808 mA h g^−1^ (404 mA h cc^−1^) from the 2nd cycle to the 100th cycle, whereas that of the bare MoO_3_ powders decreased rapidly from 1090 mA h g^−1^ (621 mA h cc^−1^) to 239 mA h g^−1^ (136 mA h cc^−1^) in the same cycle range. Additionally, the Coulombic efficiency of the MoO_3_/C composite microspheres increased quickly to above 99% after the second cycle. The amorphous C matrix of MoO_3_/C composite microspheres more effectively buffered the large volume change of the MoO_3_ active material during the fast charging–discharging process. On the other hand, the structural destruction of the bare MoO_3_ powders during repeated Li^+^-ion insertion and desertion processes resulted in capacity fading continuously. Therefore, better cycling of the MoO_3_/C composite microspheres could be achieved because of the improved structural stability of the MoO_3_.

In order to evaluate the rate performances of both samples, electrochemical tests were performed at various current densities, as shown in Figure 5d. As the current densities increased from 0.5 to 1.5, 3.0, and 5.0 A g^−1^, the MoO_3_/C composite microspheres exhibited reversible discharge capacities of 999, 875, 716, and 467 mA h g^−1^, respectively. However, the bare MoO_3_ powders delivered a low reversible discharge capacity of 352 mA h g^−1^ at 5.0 A g^−1^ as shown in Figure 5d. The C matrix of the MoO_3_/C composite microspheres improved the electrical conductivity of the sample. Additionally, the small, amorphous MoO_3_ nanograins imbedded within the C matrix decreased the diffusion distance and increased the diffusion rate of the Li^+^ ions, thus synergistically speeding up the rate of the MoO_3_/C composite microspheres more than that of the bare MoO_3_ powders.

The superior Li^+^-ion storage properties of the MoO_3_/C composite microspheres were supported by EIS analysis, as shown in Figure 6 [43,44,45]. Nyquist plots of the samples before and after cycles were obtained by deconvolution with a Randle-type equivalent-circuit model (Figure 6d). The MoO_3_/C composite microspheres and bare MoO_3_ powders had similar charge-transfer resistance (Rct) values before cycling, as shown in Figure 6a. However, the cell with the MoO_3_/C composite microspheres obtained after 100 cycles showed a lower Rct value of 42 Ω compared to that of 134 Ω for the bare MoO_3_ powders, as shown in Figure 6b,c. The structural destruction of the bare MoO_3_ powders during the repeated Li^+^-ion insertion and desertion processes increased the Rct values significantly. On the other hand, the MoO_3_ nanograins embedded within the amorphous C were not pulverized during the repeated cycles. Moreover, the C matrix served as fast and continuous transport pathways for electrons upon cycling because of its high electrical conductivity. The high structural stabilities of the MoO_3_/C composite microspheres with high lithium-ion storage capacities resulted in low Rct values during cycling. The MoO_3_/C composite microspheres with a high structural stability during repeated lithium insertion and desertion reactions showed excellent cycling and rate performance, as shown in Figure 5.

The morphologies of the MoO_3_/C composite microspheres and bare MoO_3_ powders obtained after 100 cycles are shown in Figure 7. The bare MoO_3_ powders were broken into several pieces after the cycles, as shown by the TEM image in Figure 7a. In contrast, the MoO_3_/C composite microspheres maintained their morphologies quite well even after the repeated Li^+^ insertion and desertion processes in Figure 7b,c. The excellent Li^+^-ion storage properties of the MoO_3_/C composite microspheres are, therefore, attributed to the improvement of the structural stability and electrical conductivity by the carbon composite.

## 4. Conclusions

In this study, low crystalline MoO_3_/carbon composite microspheres in which MoO_3_ nanocrystals were distributed homogeneously in the amorphous C matrix, were directly prepared by a one-step spray pyrolysis within several seconds. The MoO_3_/carbon composite was spherical, with diameters on the order of microns, because they were formed from one droplet with several tens of micrometers by a series of drying, decomposition, and crystallization processes inside the hot-wall reactor during spray pyrolysis. The amorphous C matrix of the MoO_3_/C composite microspheres effectively buffered the large volume change of the MoO_3_ active material during the fast charging–discharging process. Therefore, a better cycling of the MoO_3_/C composite microspheres could be achieved because of the improved structural stability of the MoO_3_. Additionally, the small MoO_3_ nanograins imbedded within the C matrix decreased the diffusion distance and increased the diffusion rate of Li^+^ ions, thus accelerating the rate of the MoO_3_/C composite microspheres. The superior Li^+^-ion storage properties of the MoO_3_/C composite microspheres compared to those of the bare MoO_3_ were supported by an EIS analysis and by observing the morphologies of the samples obtained after 100 cycles. The simple process introduced in this study is expected to be useful for the large-scale synthesis of TMOs/carbon composite microspheres for a wide variety of applications including energy storage.

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
