# Peer review of "Large Scale Process for Low Crystalline MoO3-Carbon Composite Microspheres Prepared by One-Step Spray Pyrolysis for Anodes in Lithium-Ion Batteries"

_nanomaterials, 2019, doi:10.3390/nano9040539_

Reviewer 1 Report

This manuscript describes to fabricate the low crystalline MoO3/carbon composite microspheres by wet synthesis for anodes in lithium-ion batteries. The composite attempt for MoO3/carbon nanoparticle by one-step spray pyrolysis technique is a unique and an available result, and the authors express that MoO3/carbon nanoparticle can obtain optimal electrochemical properties for lithium-ion secondary battery. Moreover, the authors’ original scope for the optimization of the anode active material in this study is explained well about the analytic locations of the effort among the authors’ results. However, the scientific locations for the effort should be clarified more specifically. For above reasons, this paper needs some revisions for the publication standard of this journal at the moment.

1.       Please rewrite scale bars more clearly in the SEM, TEM images of all figures.

2.       Detailed effect of processing parameters on the property of composites for LIB shall be added, and its mechanism shall be revealed in detail, especially coupled with the author’s own results, to highlight the effect of processing parameters.

3.       The authors should express the details of MoO3/carbon nanocrystal structure. How about is the interface between MoO3 and amorphous carbon matrix in a composite particle?

4.       The carbon matrix in a composite particle is a homogeneous amorphous matrix? The analysis of the synthesized carbon in a particle should be more clarified, for example, the 2D spectrum of Raman in Fig.3a and TEM/SEM image of Fig.1.

5.       The fabrication condition of the anode electrode in the study should be added clearly in your manuscript.

6.       How weight is the specific gravity of a MoO3/carbon composite? The comparison between the MoO3/carbon composite and bare MoO3 should be explained for an analysis of mAh/cc.

7.       From the cycle characteristics in Fig.5c, the capacity of the bare MoO3 degraded on around 20 cycles comparing with the MoO3/carbon composite. The authors should explain scientifically and systematically the reason of the electrochemical optimizations in Figs.5c and d with including the results of Fig.5b, Fig6 and Fig.7 in the chapters of Results and Discussions or Conclusions of your manuscript.  

Author Response

Authors would like to thank a reviewer1 for her/his time and comments on our manuscript.

[Q1] Please rewrite scale bars more clearly in the SEM, TEM images of all figures.

[A1] We highly appreciate the reviewer’s positive evaluation of our work and the reviewer’s helpful comments. As the reviewer commented, the scale bars in the SEM, TEM images were showed clearly in the revised manuscript, as shown below. 

[Q2] Detailed effect of processing parameters on the property of composites for LIB shall be added, and its mechanism shall be revealed in detail, especially coupled with the author’s own results, to highlight the effect of processing parameters.

[A2] Thanks for the reviewer’s helper comment. As the reviewer commented, the processing parameters such as reaction temperature, the flow rate of carrier gas, the ratio between Mo salt and C precursor, and the concentration of spray solution, during spray pyrolysis could effect on the final morphologies, size, and crystallinity of the obtained MoO3/carbon microspheres. Therefore, the optimum process conditions during spray pyrolysis is very important for producing powders for use in anode materials with high Li ion storage property in Li ion batteries. On these viewpoints, the authors considered the relationship between the parameters and a product and controlled the above parameters for synthesizing low crystalline MoO3/carbon composite microspheres, in which MoO3­ nanocrystals are distributed homogeneously in the amorphous C matrix, directly by one-step spray pyrolysis. Although the data of the cells assembled with powders obtained with diverse processing parameters during spray pyrolysis were not provided, the electrochemical properties of the MoO3/carbon composite microspheres were compared with the bare MoO3, in this study. However, the authors totally accept the reviewer’s helper suggestion. Additionally, the authors are going to perform experiments to confirm the effect of processing parameters on properties of the MoO3/carbon composite for LIBs, in the near future. 

[Q3] The authors should express the details of MoO3/carbon nanocrystal structure. How about is the interface between MoO3 and amorphous carbon matrix in a composite particle?

[A3] Thanks for the reviewer’s helper comment. In this study, low crystalline MoO3/C composite microspheres were prepared by spray pyrolysis for a very short reaction time of 6 seconds. During the process, PVP in a solution was decomposed to amorphous C. Concurrently, the MoO3 nanocrystals with low crystallite size were formed in C matrix by decomposition of Mo salt. In this study, MoO3 nanocrystals have very low crystallite size due to the low reaction temperature and very short reaction time of 6 seconds during the process. Unfortunately, therefore, it was difficult to observe the interface between MoO3 and amorphous carbon matrix in a composite despite performing high-resolution TEM work in Fig. 1c. However, the authors totally agree with the reviewer’s comment that it is important to confirm the interface between MoO3 and amorphous carbon matrix.

[Q4] The carbon matrix in a composite particle is a homogeneous amorphous matrix? The analysis of the synthesized carbon in a particle should be more clarified, for example, the 2D spectrum of Raman in Fig.3a and TEM/SEM image of Fig.1.

[A4] In this study, the carbon matrix of the MoO3/C composite microspheres was characterized by means of Raman spectroscopy. The degree of graphitization of the carbon material can typically be evaluated according to the intensity ratio of the D and G bands of carbon at approximately 1350 and 1590 cm-1, respectively. The peak intensity ratio between the D and G bands (ID/IG) for the MoO3/C composite microspheres was approximately 3.2, demonstrating that the carbon formed in the composite was fairly disordered. However, as the reviewer commented, it is necessary to confirm the 2D spectrum at ~2685 cm-1 of Raman. Therefore, the authors showed a Raman spectrum in the range of up to 3000 cm-1 to identify the 2D band at ~2685 cm-1. As a result, the 2D band was not observed, meaning the C matrix formed in the composite was fairly disordered amorphous carbon.

Additionally, the authors corrected the following sentence in the revised manuscript and the revised Raman spectrum was reflected in the manuscript.

“The peak intensity ratio between the D and G bands (ID/IG) for the MoO3/C composite microspheres was approximately 3.2, demonstrating that the carbon formed in the composite was fairly disordered.”

à“The peak intensity ratio between the D and G bands (ID/IG) for the MoO3/C composite microspheres was approximately 3.2 and the absence of the 2D band at ~2685 cm-1 demonstrate that the carbon formed in the composite was fairly disordered.” 

[Q5] The fabrication condition of the anode electrode in the study should be added clearly in your manuscript.

[A5] As the reviewer commented, the following sentences were added in the section of Electrochemical measurements of the revised manuscript.

Lithium cell assembly was made in Ar-filled glove box at room temperature where water and oxygen concentration was kept less than 1 ppm.”

“The working electrodes were formed by coating the slurry onto copper foils and subsequently dried at 70 oC for 3h.”

[Q6] How weight is the specific gravity of a MoO3/carbon composite? The comparison between the MoO3/carbon composite and bare MoO3 should be explained for an analysis of mAh/cc.

[A6] As the reviewer commented, electrode density and volumetric capacity of the electrode are important. Thus, calculations were made to check the exact volumetric energy density of the samples. In this work, the mass loading of MoO3/C composite and bare MoO3 powders on Cu current collector are 0.75 and 0.87 mg cm-2, respectively. Also, the thickness of the electrode was 15 μm. From these information, the densities of the synthesized MoO3/C composite and bare MoO3 powders were calculated as 0.50 and 0.57 g ml-1, respectively.

Therefore, the following sentence was corrected

“The discharge capacity of the MoO3/C composite microspheres decreased slightly from 1,066 mA h g-1 to 808 mA h g-1 from the 2nd cycle to the 100th cycle, whereas that of the bare MoO3 powders decreased rapidly from 1,090 mA h g-1 to 239 mA h g-1 in the same cycle range.”

à “The discharge capacity of the MoO3/C composite microspheres decreased slightly from 1,066 mA h g-1 (533 mA h cc-1) to 808 mA h g-1 (404 mA h cc-1) from the 2nd cycle to the 100th cycle, whereas that of the bare MoO3 powders decreased rapidly from 1,090 mA h g-1 (621 mA h cc-1) to 239 mA h g-1 (136 mA h cc-1) in the same cycle range.”

Additionally, the following Figure was added in the revised supporting information. 

[Q7] From the cycle characteristics in Fig.5c, the capacity of the bare MoO3 degraded on around 20 cycles comparing with the MoO3/carbon composite. The authors should explain scientifically and systematically the reason of the electrochemical optimizations in Figs.5c and d with including the results of Fig.5b, Fig6 and Fig.7 in the chapters of Results and Discussions or Conclusions of your manuscript.  

[A7] Thanks for the reviewer’s helper comment. The following sentences were added in the revised manuscript.

3. Results and discussion

“On the other hand, the structural destruction of the bare MoO3 powders during repeated Li+-ion insertion and desertion processes resulted capacity fading continuously.”

“The high structural stabilities the MoO3/C composite microspheres with high lithium-ion storage capacities resulted in low Rct values during cycling. The MoO3/C composite microspheres with high structural stability during repeated lithium insertion and desertion reactions showed excellent cycling and rate performance, as shown in Figure 5.”

4. Conclusions

The superior Li+-ion storage properties of the MoO3/C composite microspheres compared to those of the bare MoO3 were supported by EIS analysis and observing morphologies of the samples obtained after 100 cycles.”

Reviewer 2 Report

The manuscript reports a detailed characterization of MoO3/C material that can be used for anodes in Li-on batteries. The work is interesting, very well explained and written. The characterization of the material is complete,  the electrochemical results are exhaustively discussed. There are no critical issues in this work, which deserves to be published in the present form.

Author Response

Authors would like to thank a reviewer2 for her/his time and comments on our manuscript.

Reviewer 3 Report

The manuscript entitled Large Scale Process for Low Crystalline MoO3-Carbon Composite Microspheres Prepared by One-Step Spray Pyrolysis for Anodes in Lithium-Ion Batteries" reports a interesting work  in the world of anode electrodes.

This work is possible to be published in Nanomaterials after the following revision.
- the author should be meticulously revise the English language
and include the recent investigations on MoO3 for this kind of applications and more details about the samples preparation.

Author Response

Authors would like to thank a reviewer3 for her/his time and comments on our manuscript.

[Q1] The author should be meticulously revise the English language and include the recent investigations on MoO3 for this kind of applications and more details about the samples preparation.

[A1] We highly appreciate the reviewer’s positive evaluation of our work and the reviewer’s helpful comments. As the reviewer commented, the manuscript was revised by a qualified proofreader. Additionally, some sentences for the recent investigations on MoO3 were newly added into the Introduction part of the revised manuscript to clearly distinguish this work from the other works for the synthetic strategy of the low crystalline MoO3-C composite microspheres in lithium ion batteries.

Therefore, the following sentences were added into the Introduction part of the revised manuscript.

“Molybdenum oxides are candidate anode materials for LIBs because MoO3 exhibit good electrochemical properties, low cost, and low environmental friendly [19-21]. Therefore, MoO3 nanomaterials with diverse morphologies such as nanoparticles, hollow, nanobelts, naowiles, and porous structures have been prepared. Lee et al. [22] synthesized MoO3 nano-particles using hot filament chemical vapor deposition method (HFCVD) under an argon atmosphere. Zhao et al. [23] also synthesized MoO3 hollow microspheres by a template-free solvothermal route and subsequent heat treatment in air. The MoO3 hollow microspheres have a relatively high specific surface area. Chen et al. [24] prepared MoO3 nanobelts by hydrothermal method, in which the morphology of MoO3 nanobelts is affected with the addition of PEG. MoO3−x nanowires were prepared by Sunkara et al. [25] in a hot-filament chemical vapor deposition reactor. Ko et al. [26] prepared three-dimensional ordered macro-porous structured MoO3 by using polystyrene bead template via ultrasonic spray pyrolysis.”

[19] Whittingham, M. S. The role of ternary phased in cathode reactions. Electrochem. Soc. 1976, 123, 315-320.

[20] Meduri, P.; Clark, E.; Kim, J. H.; Dayalan, E.; Sumanasekera, G. U.; Sunkara, M. K. MoO3-x nanowire arrays as stable and high-capacity anodes for lithium ion batteries, Nano Lett. 2012, 12, 1784-1788.

[21] Xue, X. Y.; Chen, Z. H.; Xing, L. L.; Yuan, S.; Chen, Y. J. SnO2/a-MoO3 core-shell nanobelts and their extraordinarily high reversible capacity as lithium-ion battery anodes. Chem. Commun. 2011, 47, 5205-5207.

[22] Lee, S. H.; Kim, Y. H.; Deshpande, R.; Parilla, P. A.; Whitney, E.; Gillaspie, D. T.; Kim, M. J.; Mahan, A. H.; Zhang, S. B.; Dillon, A. C. Reversible lithium-ion insertion in molybdenum oxide nanoparticles. Adv. Mater. 2008, 20, 3627-3632.

[23] Zhao, X.; Cao, M.; Hu, C. Thermal oxidation synthesis hollow MoO3 microspheres and their applications in lithium storage and gas-sensing. Mater. Res. Bull. 2013, 48, 2289-2295.

[24] Mohan, V. M.; Bin, H.; Chen, W. Enhancement of electrochemical properties of MoO3 nanobelts electrode using PEG as surfactant for lithium battery. J. Solid State Electrochem. 2010, 14, 1769-1775.

[25] Meduri, P.; Clark, E.; Kim, J. H.; Dayalan, E.; Sumanasekera, G. U.; Sunkara, M. K. MoO3–x nanowire arrays as stable and high-capacity anodes for lithium ion batteries. Nano Lett. 2012, 12, 178-1788.

[26] Ko, Y. N.; Park, S. B.; Jung, K. Y.; Kang, Y. C. One-pot facile synthesis of ant-cave- structured metal oxide-carbon microballs by continuous process for use as anode materials in Li-ion batteries. Nano Lett. 2013,13, 5462–5466.

Round  2

Reviewer 1 Report

Thank you for your replies to my requests. You may correct to the text editing if possible.

Is the charge-discharge measurement condition of your prepared composites CC or CCCV? In case of loading characteristics of an anode material, the loading results are defferent between using CC and CCCV. 

Please tell us what the Rct and Rf in Fig.6 indicate in your composite.

Please correct scale bars more clearly in Fig.7. 

Author Response

Reviewer #1: Thank you for your replies to my requests. You may correct to the text editing if possible.

Reply: Authors would like to thank a reviewer1 for her/his time and comments on our manuscript. Additionally, as the reviewer commented, the manuscript was revised by a qualified proofreader.

[Q1] Is the charge-discharge measurement condition of your prepared composites CC or CCCV? In case of loading characteristics of an anode material, the loading results are different between using CC and CCCV. 

Reply: As the reviewer commented, capacity can be measured by a CC (Constant Current) or a CCCV (Constant Current - Constant Voltage). The most common way is by a CC. Therefore, in this study, MoO3/C composite microspheres and the bare MoO3 powders were measured by a CC discharge.

Therefore, the following sentence was corrected in the revised manuscript.

“The discharge/ charge characteristics of the samples were investigated by cycling over a potential range of 0.001–3.0 V at various current densities.”

à “The discharge/ charge characteristics of the samples were investigated by cycling over a potential range of 0.001–3.0 V under CC (constant-current) conditions.”

[Q2] Please tell us what the Rct and Rf in Fig.6 indicate in your composite.

Reply: Thank you for the reviewer’s helper comment. The authors newly added the fitted data of the EIS results in the revised supporting information, as shown below. 

[Q3] Please correct scale bars more clearly in Fig.7. 

Reply: Thank you for your suggestion. As the reviewer commented, the scale bars in the TEM images were showed clearly in the revised manuscript, as shown below.
